# Homomorphic Encryption-Based Federated Privacy Preservation for Deep Active Learning

**DOI:** 10.3390/e24111545

**Published:** 2022-10-27

**Authors:** Hendra Kurniawan, Masahiro Mambo

**Affiliations:** 1Graduate School of Natural Science and Technology, Kanazawa University, Kanazawa 920-1192, Japan; 2Institute of Science and Engineering, Kanazawa University, Kanazawa 920-1192, Japan

**Keywords:** privacy preserving, federated learning, active learning, homomorphic encryption

## Abstract

Active learning is a technique for maximizing performance of machine learning with minimal labeling effort and letting the machine automatically and adaptively select the most informative data for labeling. Since the labels on records may contain sensitive information, privacy-preserving mechanisms should be integrated into active learning. We propose a privacy-preservation scheme for active learning using homomorphic encryption-based federated learning. Federated learning provides distributed computation from multiple clients, and homomorphic encryption enhances the privacy preservation of user data with a strong security level. The experimental result shows that the proposed homomorphic encryption-based federated learning scheme can preserve privacy in active learning while maintaining model accuracy. Furthermore, we also provide a Deep Leakage Gradient comparison. The proposed scheme has no gradient leakage compared to the related schemes that have more than 74% gradient leakage.

## 1. Introduction

The rapid expansion of big data has propelled the development of machine learning. This tendency has presented conventional machine learning with substantial hurdles. Large data are typically held on distributed devices by different firms, and it is becoming increasingly difficult to learn a global model while resolving its associated privacy problems.

Obtaining sufficient labeled data for modeling purposes is one of the most challenging aspects of a wide variety of learning tasks, since acquiring labeled data is typically costly and requires human effort [1]. In many fields, there is an abundance of unlabeled data, and labels can be attached to such data which requires an expensive cost by the expert during the labeling process. It is possible to obtain labels in these instances, but it will be prohibitively expensive for the consumer. As far as labels are concerned, it is crucial to note that not all records are created equal.

Active learning (AL) is a technique for maximizing performance of machine learning with minimal labeling effort. In particular, it seeks to minimize labeling costs without sacrificing performance by selecting the most informative samples from an unlabeled dataset and submitting them to an oracle, e.g., a human annotator for labeling [1]. Meanwhile, the labels on records may contain sensitive information, and accessing them may incur a high query cost, e.g., obtaining permission from the relevant entity. Since traditional methods of active learning are failing to keep up with the times, privacy-preserving active learning has been hailed as a promising new technique [2].

In order to overcome the difficulties in data collection containing sensitive private information, federated learning (FL) was proposed by McMahan et al. [3]. FL allows the collaborative training of a machine learning model across several decentralized devices without data exchange. In FL, a centralized server delivers a global model to multiple distributed devices, which return local model parameters to the centralized server after training. Using the locally trained model parameters from the distributed devices, the centralized server updates the global model parameters and delivers updated global model parameters to the distributed devices. This procedure is repeated until convergence is obtained. FL has a benefit of limiting the disclosure of sensitive private data because it does not require local data sharing.

The aim of federated active learning scheme proposed in [4,5] is to provide privacy preservation for AL. This technique does not consider the data leakage from prameters, e.g., gradient, which is exchanged between the server and client during FL training to avoid data leakage. On the other hand, the use of homomorphic encryption enables a computation to be performed directly on a ciphertext to produce an encrypted result that is identical to the result calculated on the plaintext. A homomorphic encryption scheme protects parameters effectively when transferring intermediate parameters throughout the FL training process, and it has been widely used by numerous FL methods [6,7,8]. To the best of our knowledge, a homomorphic encryption-based FL scheme is not applied to provide privacy preservation for AL.

This paper studies and analyzes the privacy preservation of AL. We present a homomorphic encryption-based federated learning scheme to provide privacy preservation for AL. In summary, the main contributions of this paper are as follows:We propose FL scheme for AL with homomorphic encryption property to protect the confidentiality of the sensitive data on AL.We provide comparison and analysis of Deep Leakage Gradient (DLG) among the proposed scheme and other related schemes.

The rest of the paper is organized as follows. Section 2 describes the existing literature on privacy-preserving AL. Section 3 briefly explains the preliminary definition of active learning, federated learning, and homomorphic encryption. Section 4 provides a detailed description of the proposed scheme. The details of the datasets, experimental setup, experimental results and discussion are provided in Section 5. Section 6 discusses the vulnerability analysis of AL in FL, and finally Section 7 provides concluding remarks and future research directions.

## 2. Related Work

In this section, we explain related studies on the privacy-preserving scheme for AL. There are not many studies that comprehensively discussed the privacy-preserving AL. Previous studies in [2] provide privacy preservation using *k*-anonymity and differential privacy for AL, while the studies in [4,5,9] use a combination of FL and AL to achieve privacy preservation in active learning.

A study in privacy-preserving active learning was proposed by Feyisetan et al. [2]. They describe a method for implementing active learning with quantitative assurances that protects privacy. The authors suggest a framework for active learning that ensures the confidentiality of queries made to an external oracle. They use random probabilistic techniques to estimate if a query meets *k*-anonymity requirements. Then, after a query is assumed to satisfy *k*-anonymity, only one of *k* queries is forwarded to *n* external annotators to prevent the accumulation of privacy losses. In addition, a differential privacy technique is used in the active learning environment to pick a subset of training samples to send for annotation.

Ahn et al. proposed two approaches to preserve privacy in active learning: namely, the separated active learning (S-AL) and federated active learning (F-AL) method [5]. In S-AL, clients independently execute the AL prior to the FL. The *m*-th client uses the S-AL algorithm to its unlabeled dataset. It may appear simple because the S-AL directly uses the AL in the FL framework, including the annotation phase. In the F-AL, clients collaboratively run the AL in a distributed optimization method to select the instances that FL deems informative.

Another work related to federated learning and active learning was proposed by Goetz et al. [4]. They introduce Active Federated Learning, in which clients are selected in each round not at random but with a probability based on the existing model and client data. The scheme utilizes a value function that may be assessed on the user’s device and delivers a value to the server. The value function indicates how useful the data on that user are during each training round. Then, the server collects these evaluations and translates them into selection probabilities for the subsequent training cohort. By applying a simple value function related to the data loss suffered, the scheme can reduce the number of training cycles necessary for the model to reach a particular level of accuracy.

The active federated learning scheme by Ahmed et al. [9] focuses on analyzing the work of FL to benefit from unlabeled data, where the unlabeled data are available in each participating client. The objective of the AL phase is to obtain and label training examples from the local pool of unlabeled images. It is essential to note that the labeled training is an iterative process in which the most relevant sample from the pool is retrieved at each iteration. The maximum number of iterations is a crucial parameter, as after a certain number of iterations, the relevance of the selected sample will begin to decrease and AL will be forced to include irrelevant samples in the training set. They adopt an FL architecture inspired by the Federated Averaging (FedAvg) algorithm [3] to construct a global model by merging the stochastic gradient descent (SGD) of the local models.

## 3. Preliminary

### 3.1. Deep Active Learning

Existing research in deep learning assumes that labeled data are passive, either readily available or arbitrarily chosen to be labeled by human experts. Active labeling in deep learning seeks to obtain the greatest possible learning outcome with a restricted labeled dataset, i.e., by selecting the most pertinent unlabeled data to be labeled. The goal is to train the best classifier by selecting a subset of unlabeled data to be labeled, given a budget for the number of examples to be classified. The first work on active learning for deep learning (AL-DL) was proposed by Wang et al. [10]. AL-DL is a new active labeling approach for the cost-effective selection of data to be labeled for deep learning. AL-DL selects data using one of three metrics: least confidence, margin sampling, or entropy. The data selection strategy is applied to deep learning networks that utilize stacked restricted Boltzmann machines and stacked autoencoders.

The active labeling problem in deep learning is stated as follows. Given an unlabeled sample set $X_{U}$ and a labeled sample set $X_{L}$, the algorithm must label *n* samples from $X_{U}$ and add them to $X_{L}$ in order to minimize the classification error of a deep learning model fine-tuned by $X_{L}$. The AL-DL algorithm is based on the principle of selecting samples that are challenging to categorize by the present deep learning network. To evaluate uncertainty, an entropy calculation is employed to pick the most uncertain unlabeled samples $x_{i}$. This algorithm produces *n* unlabeled samples to be labeled. AL-DL with entropy selects the sample with the greatest entropy of class prediction information using Equation (Equation 1), where $h_{j}^{N}$ is the activation value of the unit *j* in the top layer out of *N* deep learning layers.
(1)$$x_{i}=\underset{x_{i}}{\arg\max}-\sum_{j}p\left(h_{j}^{N}|x_{i}\right)\log\;p\left(h_{j}^{N}|x_{i}\right)$$

### 3.2. Federated Learning

In this subsection, we briefly describe the federated learning (FL) that has been proposed in [3,11]. Multiple distributed devices with locally stored data train a machine learning model utilizing FL without transferring locally stored data. Distributed devices share only local model parameters produced by training a global learning model provided by a centralized server with local data, enabling them to participate in the training process without fear of data leaking. The centralized server aggregates locally learned model parameters to update the global model and distribute the updated global parameters to dispersed servers or devices for retraining. Repeat this method until convergence is obtained.

The FL model consists of the following four algorithms:Initialization takes an input of security parameter *k*, and it produces a global model $w_{G}\in R$ as an output, where R is the set of real number.*Local training* takes the global model $w_{G}\in R$, a local dataset D, and a positive integer *t* as input, and it produces a local model $w_{L}\in R$.Uploading takes the local model $w_{L}\in R$ and a positive integer *t* as input, and it generates a vector $v_{t}^{i}\in R^{N}$ as an output.Aggregation takes a set of vectors $v_{t}^{i}\in R$ and a positive integer *t* as input, and it produces the global model $w_{G}\in R$ as output.

### 3.3. Homomorphic Encryption

A homomorphic encryption scheme has a pair of algorithms with Enc function and Dec function and the following attributes:Enc function with a plaintext input $m\in Z_{N}$ outputs a ciphertext c∈C, where a ciphertext space *C* is homomorphic to the plaintext space $Z_{N}$ under Enc, i.e., $m_{1},m_{2}\in Z_{N},Enc\left(m_{1}+m_{2}\right)=Enc\left(m_{1}\right)+Enc\left(m_{2}\right)$ and/or $m_{1},m_{2}\in Z_{N},Enc\left(m_{1}m_{2}\right)=Enc\left(m_{1}\right)Enc\left(m_{2}\right)$, and identity element of $Z_{N}$ maps to identity element of *C*;Dec function with a ciphertext input c∈C outputs a plaintext $m\in Z_{N}$, where the plaintext space $Z_{N}$ and the ciphertext space *C* are homomorphic under Dec;Enc and Dec functions are computationally efficient and satisfy Dec(Enc(m))=m.

There are two forms of homomorphic encryption: additively homomorphic and multiplicatively homomorphic. Additively homomorphic encryption has a pair of Enc and Dec functions, where $m_{1}$, $m_{2}\in Z_{N}$, $c_{1}=Enc\left(m_{1}\right)$, $c_{2}=Enc\left(m_{2}\right)$, $c_{3}=c_{1}+c_{2}$, and we have $Dec\left(c_{3}\right)=m_{1}+m_{2}$. Multiplicatively homomorphic encryption has a pair of Enc and Dec functions, where $m_{1},m_{2}\in Z_{N}$, $c_{1}=Enc\left(m_{1}\right)$, $c_{2}=Enc\left(m_{2}\right)$, $c_{3}=c_{1}c_{2}$, and we have $Dec\left(c_{3}\right)=m_{1}m_{2}$.

Fully homomorphic encryption (FHE) [12] refers to a homomorphic encryption technique that works for any circuit with arithmetic and logical operations which can be efficiently evaluated through ciphertext. This atribute enables the privacy-preserving processing of sensitive data, which is a very important and currently unsatisfied demand in computing applications. Due to the performance restrictions of computer architectures, FHE techniques are not nearly ready for deployment in practical applications. Applications based on current FHE systems, which need efficient implementations of computationally expensive mathematical operations, are typically orders of magnitude slower than traditional software applications that operate on plaintext data.

Partially homomorphic encryption is a homomorphic encryption in which homomorphism is only partially supported, i.e., the encryption scheme is homomorphic for some operations but not for the other.

Somewhat homomorphic encryption is a subtype of FHE in which homomorphism is supported only for a restricted circuit, i.e., the encryption scheme is homomorphic for all operations but works only for circuits with a restricted number of operations.

A BFV homomorphic encryption scheme proposed by Fan and Vercauteren [13] extends Brakerski’s encryption technique [14] from learning with errors (LWE) to ring LWE (RLWE). The RLWE problem is merely a ring-based variant of the LWE problem. The BFV works with the following processes:Key generation algorithm takes the security parameter *k* as input and produces a public key pk and a secret key sk as output.Encryption algorithm takes a plaintext *m*, a public key pk, and a randomness *r* as input, and it produces a ciphertext *c* as output.Decryption algorithm takes a ciphertext *c* and a secret key sk as input, and it produces a plaintext *m* as output.

## 4. Proposed Scheme

In this section, we provide a detailed explanation of the privacy-preserving scheme for AL using homomorphic encryption-based FL. Figure 1 shows the proposed scheme with one server and *n* multiple clients. For the first round, the server sends the initial encrypted weight of a global model to all clients; then, clients will decrypt the weight of the global model and execute the AL training process. After completing the active learning process, clients encrypt their weight model and then share it with the server. In the second and next round, the server aggregates encrypted weight model from clients and sends the aggregated weight model back to clients.

Figure 2 describes the active learning query process of the proposed scheme using one server and one client. The server executes global model training using labeled data (U) and updates the global model (weight aggregation) using the client encrypted model as shown in Algorithm 1. Then, the client performs active learning queries to predict unlabeled instances (UI) using the decrypted global model with an input of unlabeled sample set (U) as shown in Algorithm 2. If the prediction process doesn not meet the stop criteria.

Algorithm 1 describes the procedure of global model training with labeled data samples and the calculation of the average encrypted weight of global model aggregation. In the first process, the server trains the initial global model h.fit(L) using labeled data samples and encrypts the layer.weight of the global model using $public_{\text{\_}}key$ by $Enc_{pub\text{\_}key}\left(layer.weighhts\right)$, only for the first time. The encrypted weight of the global model is shared to all clients.

In the second process, the server aggregates all the encrypted weight of local model ${\left[W\right]}_{all}$; then, it updates the weight of global model ${\left[W\right]}_{aggr}$ using the client’s encrypted weight of the local model. The average weight of the encrypted model is calculated by BFV homomorphic encryption using Equation (Equation 2), where *n* is the total number of clients {c1,c2,…,cn}, ⊗ is multiplicative homomorphic encryption, ⊕ is additive homomorphic encryption and $Enc\text{\_}Model\text{\_}Weight_{cn}$ is encrypted model weights of client *n* computed by $Enc_{pub\text{\_}key}\left(layer.weighhts\right)$. Instead of encrypting the gradient, we focus on encrypting the weights of the model. This method does not cost more in homomorphic encryption operations and the communication overhead. Further explanation is provided in Section 5.2.
(2)$$\begin{array}{c} Avg\text{\_}Enc\text{\_}Model\text{\_}Weight=\frac{1}{n}⊗\{Enc\text{\_}Model\text{\_}Weight_{c1}⊕Enc\text{\_}Model\text{\_}Weight_{c2}⊕\\ \ldots ⊕Enc\text{\_}Model\text{\_}Weight_{cn}\}, \end{array}$$

After received global model ${\left[W\right]}_{G}$ from the server, the client decrypts $Dec_{priv\text{\_}key}$ all layers of the global model ${\left[W\right]}_{G}$; then, it saves it as unencrypted local model $h_{c}$, as shown in Algorithm 2. In the next step, the client executes AL training using predicted batch_samples and unencrypted local model $h_{c}$. The AL training is executed until a stop criteria is met, and it obtains an updated local model clf. Finally, the client encrypts $Enc_{pub\text{\_}key}$ the weight of local model ${\left[W\right]}_{L}$ and shares it to the server.

Concerning the active learning operations by clients, a classifier is trained on the seed (a tiny manually labeled sample) at the first round to produce an initial model; then, the model is used to predict labels for samples in an unlabeled pool of images and add them to the seed based on parameters given in the underlying sampling strategy. It is essential to note that this is an iterative process in which the most relevant sample from the pool is retrieved at each iteration. The procedure will continue to retrieve samples from the pool until a stop condition is reached. The maximum number of iterations is a crucial parameter, as after a certain number of iterations, the relevance of the selected sample will begin to decrease, and deep AL will be forced to include irrelevant samples in the training set at some point. To achieve this objective, various ways could be employed to fix the number of iterations. One of the possible options is to terminate the procedure when the model’s accuracy reaches a steady level. Our stop criteria are determined by the maximum number of query iterations.

Since our work focus on privacy-preserving scheme utilizes deep AL, we use the image classification dataset [15,16] as a study case. The process begins with feature extraction from input images using a pre-trained deep learning model called ResNet [17]. We use ResNet trained on a dataset for image classification to extract object-level characteristics from input images. It is important to note that the feature extraction element is independent of the FL and deep AL components; hence, it is assume that the model used for feature extraction will not have a significant impact on the overall evaluation.

**Algorithm 1:** Server model training with labeled data and encrypted model aggregation

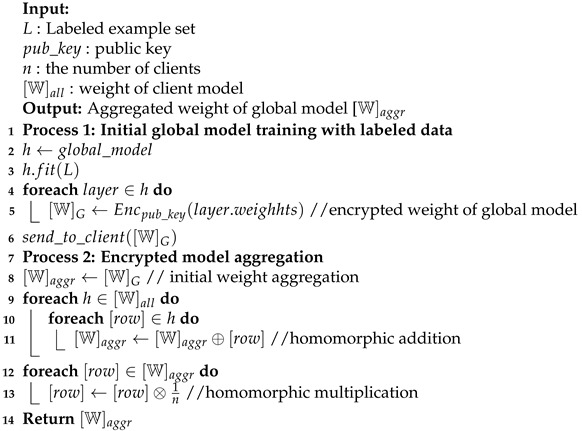



**Algorithm 2:** Active learning using unlabeled data in each client

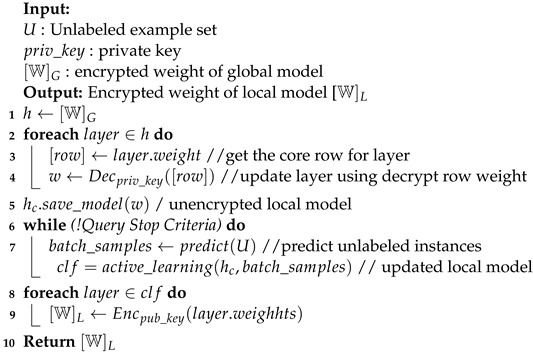



## 5. Result and Discussion

### 5.1. Datasets

The datasets for simulation are public image datasets that have been widely used in the machine learning research communities. Table 1 describes the number of samples and descriptions of each dataset. The first one is an MNIST dataset of handwritten digits formatted as 28 × 28 images, with 784 features created [15]. This dataset is composed of 60,000 examples with 10 label decisions to recognize a one-digit number between 0 and 9. The second dataset is CIFAR-10 [16], which is a well-established computer vision dataset used for object recognition. It is a collection of 60,000 tiny pictures and comprises 32 × 32 color images comprising one of ten item classes, with 6000 images per class.

### 5.2. Experimental Setup

We provide the following computing environment for the experiments. A server computer has a CPU configuration of Intel Xeon Bronze 3206R 1.90 GHz, 32.0 GB RAM, 2 TB storage and is run under a Ubuntu-20.04 operating system. The NVIDIA RTX A4000 graphics card is used for GPU computation. A Python-based ModAL framework [18] is used to provide an active learning strategy, and for Somewhat Homomorphic Encryption (SHE) computation, we use Pyfhel libraries [19]. The library offers standard SHE operations such as encoding, key generation, encryption, decryption, addition, multiplication, and relinearization.

We implement entropy-based sampling [10] and the BALD [20] strategy for active learning. The 60,000 samples of each MNIST handwritten and CIFAR-10 dataset are used during the simulation. The dataset is divided into two parts: 10,000 samples make up the labeled dataset for model training in the server, and 50,000 unlabeled data samples are shared equally by the client. Here, 1000 samples of labeled data are used for model testing at the server. A maximum of 5 clients will run 10 query rounds to predict a batch of 500 unlabeled instances and re-train the model.

A deep AL algorithm of our scheme uses Recurrent Neural Network (RNN) with long short-term memory (LSTM) [21]. LSTM is trained using the deep features extracted by ResNet. A hybrid RNN-LSTM model will improve image classification accuracy. The total number of LSTM layers is 2 with 128 hidden nodes and 64 hidden nodes in layer 1 and layer 2, respectively. We use Adam optimizer with a learning rate of 0.001. The experimental result is calculated by the average value of three simulation runs.

For each client cn, BFV homomorphic encryption is executed by $Enc\text{\_}Model\text{\_}Weight_{cn}$, as shown in Equation (Equation 2). In our implementation, the deep AL scheme has 28 × 28 features in the input layer, 128 hidden neurons in LSTM layer-1, and 64 hidden neurons in LSTM layer-2. The scheme produces 7808 weights of the local model and eight bias values. All the values are encrypted once in an array ${\left[W\right]}_{L}$ using the Pyfhel library.

To provide Avg_Enc_Model_Weight corresponding to ${\left[W\right]}_{aggr}$, our proposed scheme simply uses additive and multiplicative operations of BFV homomorphic encryption for each array of elements in ${\left[W\right]}_{L}$ by the Pyfhel library.

To set up the homomorphic encryption scheme, we apply the BFV homomorphic encryption and use pre-defined default values for the homomorphic encryption parameters in the Pyfhel library, with the exception of parameter sec. The sec parameter is used to determine the level of bit-wise security given. We conduct all the experiments with 128-bit sec parameters of BFV homomorphic encryption. The key generation and distribution are provided by the Pyfhel library, and the private key is only shared by the client to prevent the server from accessing it.

### 5.3. Classification Accuracy Performance

In this subsection, we measure the classification accuracy performance of entropy sampling and BALD deep AL strategy. We calculate two different FL schemes using both encrypted data (AL with FL-Enc) and unencrypted data (AL with FL); the classification accuracy exceeds 80% in the final round, indicating that BFV somewhat homomorphic encryption does not degrade model performance.

#### 5.3.1. Accuracy Analysis with Different Number of Clients

Table 2 provides the classification accuracy of the proposed scheme on the MNIST and CIFAR-10 datasets. We evaluate the trade-off between the number of clients and the classification accuracy. The main objective of the experiment is to determine how increasing the number of clients affects the accuracy performance. We test with the manually annotated training set for this purpose. We begin with two clients, where the total available training samples are spread across two clients, and we incrementally expand the number of clients. As the number of clients increases, the classification accuracy decreases, resulting in a not-so-large reduction in the number of training samples for each client. Comparing the FL scheme using both encrypted data (AL with FL-Enc) and unencrypted data (AL with FL), the encrypted data have lower accuracy, but both FL schemes are still comparable.

We compare the classification accuracy for the MNIST and CIFAR-10 dataset for both entropy sampling and BALD AL strategy. The accuracy of FL scheme with encrypted data (AL with FL-Enc) is lower compared to the FL scheme with unencrypted data (AL with FL). This is due to the effect of homomorphic encryption, but the decrease in accuracy is not significant and still tolerable.

#### 5.3.2. Accuracy Analysis as a Function of Query Rounds

We provide accuracy analysis as a function of query rounds for a scheme of one server and two clients on the MNIST dataset as shown in Figure 3. For the first round, entropy sampling has the accuracy of 0.7334 and 0.7312 for scheme of AL with FL and AL with FL-Enc, respectively. BALD AL with FL has 0.8403 accuracy and BALD AL with FL-Enc has 0.8352 accuracy. With the increasing number of query rounds, the batch sample also increases, and the classification accuracy also increases. Finally, at round 10, the entropy sampling reaches 0.9072 and 0.9023 for AL with FL and AL with FL-Enc. BALD AL with FL has 0.9289 accuracy, and BALD AL with FL-Enc has 0.9221 accuracy.

### 5.4. Execution Time Performance

The next experimental result is provided in Table 3. We evaluate the execution time (second) of entropy sampling and BALD AL strategy using both unencrypted data (AL with FL) and encrypted data (AL with FL-Enc). The experimental result shows in both the MNIST and CIFAR-10 datasets; there is a significant difference of execution time for the FL scheme on unencrypted and encypted data. The FL scheme with encrypted data is seven to ten time slower compared to the FL scheme with unencrypted data.

In common machine learning applications, the training phase is usually not performed in real time. Only the inference phase of the final model is executed in real-time application: for example, the machine learning application on the object detection of edge devices with CCTV cameras. The model is trained using adequate video files that have been stored in the cloud/server for a certain time to obtain better accuracy. The final model will be installed in the edge device to perform machine-learning object detection using real-time video files from the CCTV camera. In the proposed scheme, the final model aggregated by the server can be used by the client to perform real-time machine-learning applications. Even though the proposed scheme uses homomorphic encryption to compute the final model, the delay does not occur in the inference phase.

The benefits we obtain from our proposed scheme are: (1) the server can update its model based on the encrypted model of the client; (2) ensure there is no privacy breach, since the user data are kept at their location/device and only the encrypted model is shared to the server.

## 6. Vulnerability Analysis

FL presents a new paradigm for protecting user privacy while executing large-scale machine learning activities, but it is riddled with vulnerabilities that must be handled. Knowledge of FL vulnerabilities helps to keep track and defend against potential assaults. The inability to detect FL vulnerabilities will affect defenses that are prone to attack. We provide source vulnerability analysis for the proposed scheme and compare it to related work schemes.

In the scheme of Ahn et al. [5], a cross-silo FL consisting of a server and numerous clients is analyzed. The FedSGD [3] is used for FL updates, where the global model is produced by iterative stochastic gradient descent (SGD). Additionally, the FedAvg [3] computes the converged solution at each client by iterating numerous times prior to calculating the average. Related work by Ahmed et al. [9] also uses Federated Averaging (FedAvg) algorithm [3] to construct a global model by merging the stochastic gradient descent (SGD) of the local models.

In the scheme of Goetz et al., a subset of users is selected during each training iteration. Each user trains the model using their own data and generates updated model parameter values. These updated model parameter values are then sent to the server and aggregated using Federated ADAM [22]. This approach aims to pick an optimum subset of users based on a value function that reflects the usefulness of each user’s data throughout each training round. A differentially private mechanism [23] is used to protect the value function during transmission.

### 6.1. Source Vulnerability Analysis

Bouacida and Mohapatra [24] provide nine categories of source vulnerabilities. Table 4 shows a comparison of source vulnerability and possible attacks to the proposed scheme and related schemes.

Communication: The annotation process is usually executed locally in a client for the AL scheme, but multiple communication cycles between the server and clients are required for exchanging models in FL. An insecure channel represents an exposed vulnerability. The models shared between participants and the final FL model in the deployment phase can be intercepted and replaced with malicious models by eavesdroppers. All communication channels are insecure in both the proposed scheme and related schemes; there is a possibility that an adversary can intercept and change the original models with malicious ones.Gradient Leakage: FL provides a technique that protects privacy when training with distributed data. Despite the fact that the data are not explicitly shared throughout the training phase, it is still possible for adversaries to expose sensitive information and even resemble the raw data by sharing the gradients. The proposed scheme uses homomorphic encryption-based FL. The weights of the gradient are shared in encrypted mode, so there is no gradient leakage for the proposed scheme. Goetz et al. [4] add noise to the model using differential privacy. Still, there is a gradient leakage in the scheme. Both Ahn et al. [5] and Ahmed et al. [9] do not have any additional method to the scheme, so there is a gradient leakage in the scheme.Compromised Clients: Clients are regarded as a crucial component of the AL in the FL scheme. Compromised clients distort the FL training process by using model parameters or training data to create an attack. All schemes are highly vulnerable to attacks by compromised clients.Compromised Server: The server is responsible for distributing the initial model parameters, aggregating model updates, and sending the global model to the clients. The server is susceptible to some attacks such as Denial of Service (DDos) attacks. With the current server conditions, all schemes are susceptible to attacks carried out on servers to affect processes of a model in FL.Aggregation Algorithm: The inadequate configuration and maintenance of a strong aggregation technique will leave the global model vulnerable and unreliable. The proposed scheme has advantages over the related schemes because it has a homomorphic encryption configuration of the aggregation algorithm.Non-Malicious Failure: Particular clients will report failures and, as a result, will drop out of the training cycle. Such failures may lead to the elimination of clients with relevant training data, resulting in a low-quality, biased model. Clients on each scheme can cause non-malicious failure, because there is no known protocol used to solve this problem both in the proposed scheme and the related schemes.Distributed Nature of FL: Distributed training facilitates collusion and distributed attacks, in which numerous participants collude to conduct an organized attack. Similar to the criteria for compromised clients, all schemes have limitations to overcome the problems in the distributed nature of FL criteria.FL Environment Scope: The FL scheme involves numerous parties, such as clients, architects, developers, analysts, and deployers. Designing and implementing coordination rules can be challenging and may result in instances where the robustness of collaborative training cannot be guaranteed. In all schemes, both the proposed scheme and the related schemes do not have a protocol in coordinating various FL elements to ensure the collaborative training.Model Deployment: An adversary could interfere with the training procedure to generate or aggravate inference-time flaws in the deployed model. All schemes are at risk of enemies that can interfere with the training process, but the proposed scheme has a better approach by using homomorphic encryption on the model deployment.

### 6.2. Gradient Leakage Analysis

Exchanging gradients is a widely used method in a modern distributed machine learning system (e.g., federated learning). Zhu et al. [25] present an approach which shows the possibility of obtaining private training data from publicly shared gradients, namely Deep Leakage Gradient (DLG). They synthesize the dummy data and corresponding labels with the supervision of shared gradients. An improvement of DLG is presented by Zhao et al. [26]. They provide an analytical approach to extract the ground-truth labels from the shared gradients, namely improved Deep Leakage from Gradients (iDLG). The iDLG is capable of extracting the data more effectively based on correct labels.

We provide an experiment of iDLG on the proposed scheme and related schemes for the MNIST dataset. We run all experiments 100 times by following the settings in [26]. Table 5 shows the comparison of accuracy of the extracted labels for iDLG. Since there is no additional improvement on the gradient-sharing mechanism of Ahn et al. [5] and Ahmed et al. [9], the iDLG can extract almost all labels of the MNIST image with 98% accuracy. The iDLG result on the scheme of Goetz et al. [4] has 74% accuracy because the scheme has an additional procedure using the differential private mechanism to put noise in their scheme. The result of iDLG on the proposed scheme without encryption also has high accuracy with 98%. The significant result shows there is no gradient leakage for the proposed scheme with encryption. The iDLG can not extract the label because of the homomorphic encryption scheme used in FL.

## 7. Conclusions

In this paper, we presented a privacy-preserving federated learning (FL) scheme to protect the privacy of user data for deep active learning (AL). The homomorphic encryption scheme used in it can protect the weight of the deep AL model. A detailed evaluation of two different AL methods, namely entropy-based and BALD scheme, have been provided. We demonstrated that FL could be advantageous for deep AL that lacks large-scale annotated datasets. In addition, we analyzed the impact of multiple clients architecture on the performance of the encrypted global model. The experimental result shows that the proposed homomorphic encryption-based FL can preserve privacy for deep AL while keeping the accuracy, and the scheme has no gradient leakage. In the future, we aim to provide an improvement of the proposed scheme by considering the specific attack to federated learning and utilizing an optimized homomorphic encryption scheme.

## Figures and Tables

**Figure 1 entropy-24-01545-f001:**
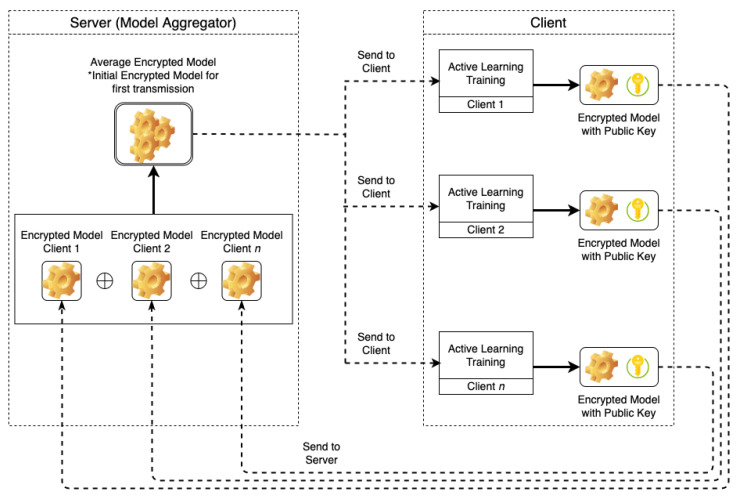
Proposed scheme of homomorphic encryption-based federated active learning with one server and multiple clients.

**Figure 2 entropy-24-01545-f002:**
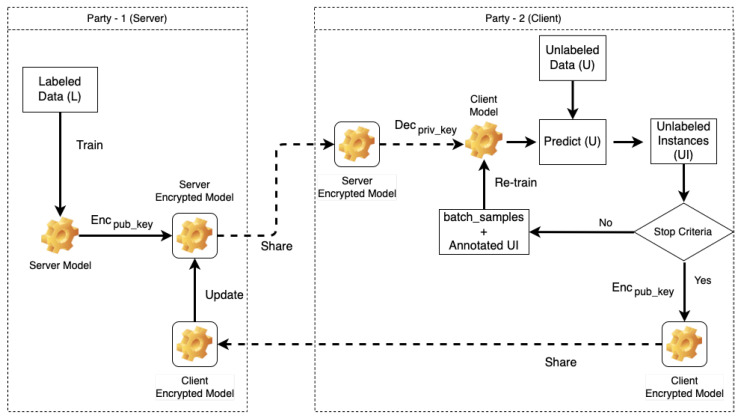
Active learning query process with one server and one client of the proposed scheme.

**Figure 3 entropy-24-01545-f003:**
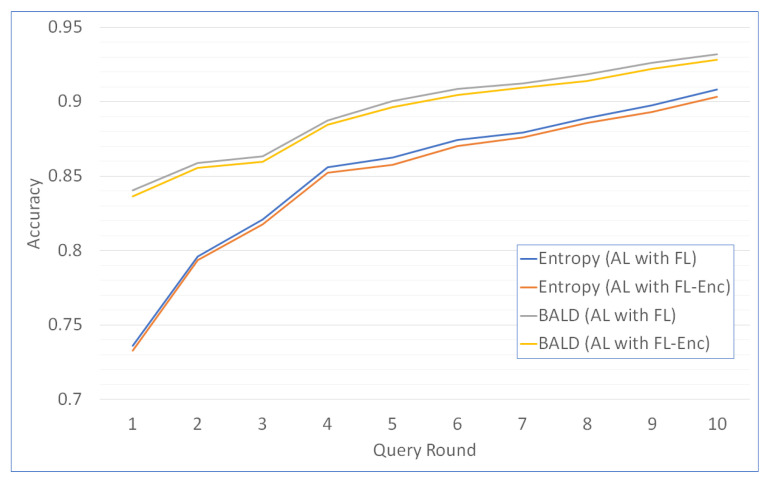
MNIST classification accuracy of Active Federated Learning as a function of query rounds for two clients. (AL: Active Learning, FL: Federated Learning, FL-Enc: FL with homomorphic encryption).

**Table 1 entropy-24-01545-t001:** Public dataset used in simulation.

No	Dataset	Samples	Features	Classes
1	MNIST	60,000	784	10
2	CIFAR-10	60,000	1024	10

**Table 2 entropy-24-01545-t002:** Accuracy of Active Federated Learning after 10 query rounds with multiple clients. (AL: Active Learning, FL: Federated Learning, FL-Enc: FL with homomorphic encryption).

Dataset	No of Clients	Entropy Accuracy		BALD Accuracy	
		AL with FL	AL with FL-Enc	AL with FL	AL with FL-Enc
MNIST	2	0.9072	0.9023	0.9312	0.9262
MNIST	3	0.9023	0.9008	0.9287	0.9231
MNIST	4	0.8971	0.8965	0.9176	0.9082
MNIST	5	0.8942	0.8912	0.9120	0.8988
CIFAR-10	2	0.8872	0.8829	0.9156	0.9093
CIFAR-10	3	0.8845	0.8789	0.9086	0.8965
CIFAR-10	4	0.8763	0.8651	0.8972	0.8905
CIFAR-10	5	0.8591	0.8522	0.8924	0.8878

**Table 3 entropy-24-01545-t003:** Execution Time of Federated Active Learning after 10 query rounds with multiple clients. (AL: Active Learning, FL: Federated Learning, FL-Enc: FL with homomorphic encryption).

Dataset	No of Clients	Entropy Time (Second)		BALD Time (Second)	
		AL with FL	AL with FL-Enc	AL with FL	AL with FL-Enc
MNIST	2	218	1596	647	4721
MNIST	3	238	1887	704	5582
MNIST	4	265	2496	784	7384
MNIST	5	349	3397	1032	10,049
CIFAR-10	2	327	2463	1073	7232
CIFAR-10	3	356	2826	1169	9262
CIFAR-10	4	397	3738	1301	12,249
CIFAR-10	5	523	5088	1713	16,671

**Table 4 entropy-24-01545-t004:** Comparison of attack possibility from sources of vulnerabilities for the proposed scheme and related schemes. Y and N indicate the attack can be performed and cannot be performed from the source of vulnerability, respectively.

Source of Vulnerability	Ahn et al. [5]	Goetz et al. [4]	Ahmed et al. [9]	AL with FL	Proposed AL with FL-Enc

1. Communication	Y	Y	Y	Y	Y
2. Gradient Leakage	Y	Y	Y	Y	N
3. Compromised Clients	Y	Y	Y	Y	Y
4. Compromised Server	Y	Y	Y	Y	Y
5. Aggregation Algorithm	Y	Y	Y	Y	N
6. Non-Malicious Failure	Y	Y	Y	Y	Y
7. Distributed Nature of FL	Y	Y	Y	Y	Y
8. FL Environment Scope	Y	Y	Y	Y	Y
9. Model Deployment	Y	Y	Y	Y	N

**Table 5 entropy-24-01545-t005:** Comparison of improved Deep Leakage Gradient accuracy of extracted labels for proposed scheme and related works. In this comparison, scheme with lower accuracy is better.

No	Scheme	Accuracy of the Extracted Labels for iDLG
1	Ahn et al. [5]	98%
2	Goetz et al. [4]	74%
3	Ahmed et al. [9]	98%
4	Proposed AL with FL	98%
5	Proposed AL with FL-Enc	0%

## Data Availability

Not applicable.

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
