# Peer review of "Homomorphic Encryption-Based Federated Privacy Preservation for Deep Active Learning"

_entropy, 2022, doi:10.3390/e24111545_

Round 1

Reviewer 1 Report

This paper discusses active learning as a technique for investigating a way to maximize performance with minimal labelling effort. The authors propose a homomorphic encryption-based federated learning approach to protect privacy in active earning, taking the advantages of federated learning design to provide privacy preservation of user data, and the strong security level of homomorphic encryption.

The paper reads well and includes evidence of evaluation. However, there is an area for improvement, for example:

It is recommended that the abstract includes more results rather than confirming the proposed model works. E.g., could you quantify and share the impact of the privacy-enhancing technology used on usability?

The related work section requires more work and it will be useful to avoid strong statements such as "no related work establishes encryption-based FL in AL and no works investigates the effect of AL in FL", what is your evidence for this claim, e.g., did you find a systematic review paper to confirm Although it is not fully explained how you define the "effect", I can easily see many papers covering this concept such as one that aims to evaluate the benefits of active learning in a federated learning (L. Ahmed, K. Ahmad, N. Said, B. Qolomany, J. Qadir and A. Al-Fuqaha, "Active Learning Based Federated Learning for Waste and Natural Disaster Image Classification," in IEEE Access, vol. 8, pp. 208518-208531, 2020, doi: 10.1109/ACCESS.2020.3038676.)

Nonetheless, many other studies in this area and it is recommended to include more within your study,  e.g.,

Runhua Xu, Nathalie Baracaldo, Yi Zhou, Ali Anwar, and Heiko Ludwig. 2019. HybridAlpha: An Efficient Approach for Privacy-Preserving Federated Learning. In Proceedings of the 12th ACM Workshop on Artificial Intelligence and Security (AISec'19). Association for Computing Machinery, New York, NY, USA, 13–23. https://doi.org/10.1145/3338501.3357371

Ahmadi-Assalemi, G., Al-Khateeb, H., & Aggoun, A. (2022). Privacy-enhancing technologies in the design of digital twins for smart cities. Network Security, 2022(7).

Several sections in your paper teach the content, I can still see some value to clarify the preliminaries. However, you require more analysis of related work too to establish the means to also support your evaluation section. There is evidence that your model works, but how else does it compare to efforts from related work? 

Author Response

I attach the reply file to the reviewer report.

Reviewer 2 Report

The following should be noted and corrected accordingly:

1. How practicable is your proposed model in real-time? 

2. Is it cost-efficient?

3. Some diagrams and terms are not properly explained.

4. Grammar is not up to standard and requires extensive re-editing

5. Are the formulas and numbers here generic or generated by you?

6. The Related Work section is too brief and needs to be more comprehensive 

Study and consider the following related papers to embellish your paper:

• https://doi.org/10.1111/exsy.13072

• https://doi.org/10.3390/s22155574

• https://doi.org/10.1007/s00779-021-01607-3

Major revisions are required.

Author Response

(The authors gave the same response as above.)

Reviewer 3 Report

Its an interesting paper on how to execute federated deep learning protecting potentially sensitive data in the labels/annotations. Give current importance of data and data-privacy and its impact on AI/DL research this is a very timely and well-executed piece of research. It is thorough & comprehensive.  

Author Response

(The authors gave the same response as above.)

Round 2

Reviewer 1 Report

Most requirements have been addressed to a satisfactory level e.g., related work. Good to see further improvement in the abstract too to report results. e.g., "we also provide a Deep Leakage Gradient 9 comparison." and  "The proposed scheme has no gradient leakage compared to the related schemes that 10 have more than 74% of gradient leakage."

Reviewer 2 Report

The following should be noted and corrected accordingly:

1. What are the limitations of the study?

2. More references need to be added in the Related Work section 

3. Study and consider the following related paper to embellish your paper:

• https://doi.org/10.1111/exsy.13072